# New Perspectives for Low Muscle Mass Quantity/Quality Assessment in Probable Sarcopenic Older Adults: An Exploratory Analysis Study

**DOI:** 10.3390/nu16101496

**Published:** 2024-05-15

**Authors:** Maria Besora-Moreno, Elisabet Llauradó, Claudia Jiménez-ten Hoevel, Cristina Sepúlveda, Judit Queral, Glòria Bernal, Laura Pérez-Merino, Sergio Martinez-Hervas, Blanca Alabadi, Yolanda Ortega, Rosa Maria Valls, Rosa Solà, Anna Pedret

**Affiliations:** 1Functional Nutrition, Oxidation, and Cardiovascular Diseases Group (NFOC-Salut), Facultat de Medicina i Ciències de la Salut, Universitat Rovira i Virgili, 43201 Reus, Spain; mariadelaserra.besora@urv.cat (M.B.-M.); claudia.jimenez@iispv.cat (C.J.-t.H.); cristina.sepulveda@fundacio.urv.cat (C.S.); judit.queral@urv.cat (J.Q.); rosamaria.valls@urv.cat (R.M.V.); anna.pedret@urv.cat (A.P.); 2Institut Investigació Sanitària Pere i Virgili (ISPV), 43204 Reus, Spain; 3Hospital Universitari Sant Joan de Reus, 43204 Reus, Spain; gloria.bernal@salutsantjoan.cat (G.B.); laura.perez@salutsantjoan.cat (L.P.-M.); 4Service of Endocrinology and Nutrition, Hospital Clínico Universitario of Valencia, 46010 Valencia, Spain; sergio.martinez@uv.es (S.M.-H.); balabadi@incliva.es (B.A.); 5Department of Medicine, University of Valencia, 46010 Valencia, Spain; 6INCLIVA Biomedical Research Institute, 46010 Valencia, Spain; 7CIBER de Diabetes y Enfermedades Metabólicas Asociadas (CIBERDEM), 28029 Madrid, Spain; 8Institut Universitari d’Investigació en Atenció Primària-IDIAP Jordi Gol, 43202 Tarragona, Spain; yolanda.ortega@urv.cat; 9Primary Care Centre Salou, Institut Català de la Salut, 43840 Tarragona, Spain

**Keywords:** sarcopenia, muscle mass, ultrasound, isokinetic, bioelectrical impedance, older adults

## Abstract

Background: Low muscle mass quantity/quality is needed to confirm sarcopenia diagnosis; however, no validated cut-off points exist. This study aimed to determine the diagnostic accuracy of sarcopenia through muscle mass quantity/quality parameters, using the bioimpedance analysis (BIA), isokinetic, and ultrasound tools in probable sarcopenic community-dwelling older adults (≥60 years). Also, it aimed to suggest possible new cut-off points to confirm sarcopenia diagnosis. Methods: A cross-sectional exploratory analysis study was performed with probable sarcopenic and non-sarcopenic older adults. BIA, isokinetic, and ultrasound parameters were evaluated. The protocol was registered on ClinicalTrials.gov (NCT05485402). Results: A total of 50 individuals were included, 38 with probable sarcopenia (69.63 ± 4.14 years; 7 men and 31 women) and 12 non-sarcopenic (67.58 ± 4.54 years; 7 men and 5 women). The phase angle (cut-off: 5.10° men, *p* = 0.003; 4.95° women, *p* < 0.001), peak torque (cut-off: 66.75 Newtons-meters (N-m) men, *p* < 0.001; 48.35 N-m women, *p* < 0.001), total work (cut-off: 64.00 Joules (J) men, *p* = 0.007; 54.70 J women, *p* = 0.001), and mean power (cut-off: 87.8 Watts (W) men, *p* = 0.003; 48.95 W women, *p* = 0.008) in leg extension, as well as the the forearm muscle thickness (cut-off: 1.41 cm (cm) men, *p* = 0.017; 0.94 cm women, *p* = 0.041), had great diagnostic accuracy in both sexes. Conclusions: The phase angle, peak torque, total work, and mean power in leg extension, as well as forearm muscle thickness, had great diagnostic accuracy in regard to sarcopenia, and the suggested cut-off points could lead to the confirmation of sarcopenia diagnosis, but more studies are needed to confirm this.

## 1. Introduction

Currently, according to the revised European consensus on the definition and diagnosis of sarcopenia from the European Working Group on Sarcopenia in Older People (EWGSOP2), probable sarcopenia is defined as low muscle strength, but the presence of low muscle mass (either quantity or quality) is necessary to confirm the diagnose of sarcopenia [1]. In addition, sarcopenia is considered severe when low physical performance is added to low muscle strength and low muscle mass [1]. However, the previous consensus on sarcopenia (EWGSOP1) defined presarcopenia as low muscle mass, and if this was accompanied by low muscle strength or low physical performance, the diagnosis of sarcopenia was confirmed [2]. 

A key point is the differences between criteria and cut-off points proposed in EWGSOP1 and EWGSOP2 [3]. The EWGSOP1 cut-off points were less restrictive than those of the EWGSOP2 consensus; therefore, the prevalence of sarcopenia in the population is lower using EWGSOP2 criteria [3]. This fact could suggest that the EWGSOP2 criteria could be underdiagnosing sarcopenia [4]. Sometimes, older adults complain about their physical performance or strength but, according to the current diagnostic criteria, they do not have a diagnosis of sarcopenia [5].

The gold-standard techniques to assess muscle mass are magnetic resonance imaging and computed tomography [1]. However, regarding the quantity of muscle mass, bioelectrical impedance analysis (BIA) is a more affordable and portable tool to be used in clinical practice [1,2]. Muscle mass quantity determined by BIA is assessed as appendicular skeletal muscle mass (ASM; kilograms (kg)) or appendicular skeletal muscle mass index (ASMI (ASM/height^2^); kg/m^2^) based on the EWGSOP2 [1]. However, according to EWGSOP1, muscle mass was assessed as skeletal muscle mass (SM; kg) and skeletal muscle mass index (SMI (SM/height^2^); kg/m^2^) [2]. 

Regarding the quality of muscle mass, it is defined as changes in the morphology, composition, and structure of the muscle, but it also includes muscle function, which is defined as the ratio between muscle strength or power per unit of muscle mass or area [6]. The phase angle obtained by a multifrequency BIA that measures resistance and reactance to electrical current provides information about muscle quality [1,7]. Furthermore, there are other tools to assess the quality of the muscle mass, such as isokinetic and ultrasound techniques [6,8]. The isokinetic assessment tool, apart from assessing the muscle strength of lower limbs, is focused on muscle function measured by an isokinetic dynamometer [9], but its use in clinical practice is complicated due to the specificity of the equipment, the economic cost, and the high training required [1,2]. Instead, ultrasound is a valid, reliable, affordable, and non-invasive assessment tool [10] that assess both muscle mass quality, in particular, muscle function and morphology (muscle composition, structure, and stiffness) [6,8], and muscle mass quantity in different muscle groups [8,11]. The most used muscle groups for ultrasound assessment are large muscle groups, such as quadriceps, because of their accessibility and size [8]. However, smaller muscle groups, such as the forearm, could also be useful [8].

Unfortunately, unlike muscle mass quantity, currently, no validated cut-off points for muscle mass quality parameters are defined [6]. In this context, there is a need to identify new valid and clear cut-off points for sarcopenia diagnosis, considering that gold-standard techniques such as magnetic resonance imaging and computed tomography are not feasible to use in clinical practice [12,13]. Thus, considering the current limitations for the sarcopenia diagnosis and highlighting that sarcopenia could be underdiagnosed in community-dwelling older adults according to the current criteria, the identification of new cut-off points for the muscle mass quantity and quality parameters and the incorporation of them in the sarcopenia assessment could improve the sarcopenia diagnosis in the clinical practice.

This study aimed to determine the diagnostic accuracy of sarcopenia by muscle mass quantity and quality parameters using the BIA, isokinetic, and ultrasound tools in probable sarcopenic community-dwelling older adults (≥60 years). Also, we aimed to suggest possible new cut-off points for muscle mass quantity and quality parameters to confirm sarcopenia diagnosis.

## 2. Materials and Methods

### 2.1. Study Design

A cross-sectional exploratory analysis study was conducted between September 2022 and July 2023 from Reus and its outskirts (Spain), and all study visits took place at the Centre Mèdic Quirúrgic (CMQ) of Reus. The cross-sectional study was performed with two different populations, community-dwelling older adults with probable sarcopenia and non-sarcopenic ones. The present study was conducted in accordance with the Helsinki Declaration and Good Clinical Practice Guidelines of the International Conference of Harmonization (GCP ICH) and STROBE criteria [14]. The study was approved on 31 March 2022 by the Drug Research Ethics Committee (Comité de Ética de Investigación con medicamentos—CEIm) (033/2022), and the protocol was registered on ClinicalTrials.gov (NCT05485402). Also, all volunteers gave their informed consent before their inclusion in the study.

### 2.2. Eligibility Criteria

The inclusion criteria were (a) men and women between ≥60 years and ≤80 years of age, with written informed consent before the initial visit; (b) community-dwelling older adults with low muscle strength categorized as probable sarcopenia based on handgrip strength (<30 kg men and <20 kg women); and (c) community-dwelling older adults with higher muscle strength categorized as non-sarcopenic older adults based on handgrip strength (>35 kg men and >25 kg women). We used the EWGSOP1 criteria [2] to assess muscle strength because cut-off points are less restrictive than those of the EWGSOP2 consensus [1]. However, the criteria of probable sarcopenia were maintained from the EWGSOP2 because all the older adults included in the present cross-sectional study had only the muscle strength altered, the parameter that the EWGSOP2 consensus determines to deteriorate first [1].

The exclusion criteria were as follows: (a) type 1 or type 2 diabetes; (b) anemia (hemoglobin ≤13 g/deciliter (g/dL) in men, and ≤12 g/dL in women); (c) intestinal malabsorption diseases; (d) fructose and/or sucrose intolerance; (e) malnutrition (albumin <3.5 g/dL); (f) renal diseases; (g) chronic alcoholism; (h) current or past participation in a clinical trial or consumption of a research product in the 30 days before inclusion in the study; (i) institutionalized elderly; and (j) failure to follow the study guidelines. 

### 2.3. Assessment of Sarcopenia Parameters

Skeletal muscle strength based on handgrip strength (HGS) was measured twice using a validated hydraulic handheld dynamometer (JAMAR^®^ Plus+ Dynamometer; Performance Health Supply, Inc., Cedarburg, WI, USA) [1]. It was measured twice, and the average of the two values was calculated. 

Muscle mass was assessed via the BIA, using a segmental multifrequency body composition analyzer (TANITA MC-780MA; Tanita Corp., Tokyo, Japan). The muscle mass quantity parameters assessed were the SM (kg), SMI (kilograms/meter^2^; kg/m^2^), ASM (kg), ASMI (kg/m^2^), and phase angle (°). Also, the phase angle (°) assessed by BIA reported information about the quality of muscle mass. The cut-off points to determine low muscle mass parameters were <8.87 kg/m^2^ of SMI in men and <6.42 kg/m^2^ of SMI in women based on the EWGSOP1 consensus [2]; and <20 kg of ASM in men and <15 kg of ASM in women, and <7 kg/m^2^ in men and <5.5 kg/m^2^ in women based on EWGSOP2 consensus [1]. 

Also, physical performance was measured twice by 4 m gait speed, and the average of the two values was calculated [1]. A gait speed of ≤0.8 m/s (m/s) was considered low physical performance [1,2].

Other body composition parameters, such as body weight and body mass index (BMI), were assessed via the BIA. Also, height was assessed using a wall-mounted stadiometer (Tanita Leicester Portable; Tanita Corp., Barcelona, Spain).

### 2.4. Assessment of Isokinetic Parameters

The isokinetic assessment was carried out with an isokinetic dynamometer (Biodex System 4; Biodex Medical Systems, New York, NY, USA). The test was performed on the dominant leg by one repetition at an angular velocity of 180° s^−1^ in extension and flexion [15]. The volunteer was in a seated position, with the hips flexed at 90° restricted to extension/flexion of 0 to −90, with a break of 2 min between sets. The variables at 180° s^−1^ were as follows: maximum peak torque (Newtons-meters; N-m), defined as the maximum force produced by the musculature; maximum total work (Joules; J), defined as the workload at a defined angular velocity; and mean power (Watts; W), defined as the total work during a specified period [16].

### 2.5. Assessment of Ultrasound Parameters

Ultrasound muscle mass assessment was evaluated at the upper leg (quadriceps femoris (QF) and rectus femoris (RF)) and at the forearm (FA). Ultrasound assessment was conducted by VINNO 5 (Vinno (Suzhou) Co., Ltd., Suzhou, China) in the HAR-mode, with the MSK superficial preset at a frequency of 10 MHz, with a linear transducer. All ultrasound measures were carried out three times by a single researcher, and the mean of the three values was calculated for analysis.

The upper-leg ultrasound was evaluated on the dominant leg, with the volunteer in a supine position and full extension [8,17]. Measurements were taken between the anterior super iliac spine and the proximal end of the patella at 50% and 30% proximal of the superior border of the patella, with the transducer placed perpendicular to the length of the leg (Appendix A Figure A1) [8,17]. The muscle parameters assessed were subcutaneous fat thickness (SFT; centimeter (cm)) (at 30% and 50%), QF-muscle thickness (MT; cm) (at 30% and 50%), RF-MT (at 30% and 50%; cm), RF-width (W; cm) (at 30%), RF-cross-sectional area (CSA; cm^2^) (at 30%), and RF-perimeter (P; cm) (at 30%) [8,17]. Additionally, another muscle parameter assessed was the pennation angle (°) 30–50% (to get better image quality) between the anterior super iliac spine and the proximal end of the patella, with transduce parallel to the long axis (Figure 1) [8]. All of these parameters assessed muscle quantity, except for the pennation angle, which assessed muscle quality, and the CSA, which assessed both muscle quantity and quality [6].

The forearm ultrasound was assessed using the dominant arm, with the volunteers in a sitting position in a chair without armrests, with feet completely on the floor in the following position: hips and knees at approximately 90°, 30–45° shoulder flexion, 45° elbow flexion, 15–30° wrist extension, and 0–15° ulnar deviation [18]. A ball that was tennis-ball size was located in the volunteer’s hand to maintain the position and to achieve a mid-pronation/supination placement of the arm [18]. The measurement point was at 30% proximal between the styloid process of the radius and the insertion of the biceps brachii muscle into the radial tuberosity (Figure 1) [18]. The FA-MT (cm) measure included the distance from the subcutaneous adipose tissue–muscle to the muscle–bone interfaces [18].

### 2.6. Statistical Methods

Categorical variables were presented as percentages (%). Continuous variables with normal distribution were presented as the mean ± standard deviation (SD), and variables with non-normal distribution were presented as the median and interquartile range. The normality of the variables was assessed by the Kolmogorov–Smirnov test. The *t*-test and Mann–Whitney U test were used to analyze changes, depending on the nature of the variable. Moreover, the univariate general linear model was used to assess the differences among variables, adjusted by sex. Also, the effect size of the analysis was calculated.

A ROC analysis was performed to assess the diagnostic accuracy of BIA, isokinetic, and ultrasound assessment tools in probable sarcopenic community-dwelling older adults, using the area under the curve (AUC) and 95% confidence interval (CI). The ROC analysis was significant when the AUC was >0.5 and the lower 95% CI value was >0.5 (reference line = 0.5) [19]. The diagnostic accuracy based on the AUC was as follows: ≥0.9 AUC, excellent; 0.80–0.89 AUC, good; 0.70–0.79 AUC, fair; 0.60–0.69 AUC, poor; and 0.50–0.59 AUC, fail [19]. The Youden’s index was used to determine the cut-off point for each variable based on sensibility and specificity to confirm the sarcopenia diagnosis. The higher Youden’s index determined the best cut-off point [19]. 

All statistical analyses were performed using SPSS IBM (Corp. Released 2023. IBM SPSS Statistics for Windows, Version 29.0.1.0 Armonk, NY, USA: IBM Corp.). A *p*-value of <0.05 was considered significant.

## 3. Results

A total of 50 older adults were included in the cross-sectional study; 38 were older adults with probable sarcopenia with low muscle strength (69.63 ± 4.14 years) and 12 were non-sarcopenic older adults with any sarcopenia parameters altered (67.58 ± 4.54 years). Table 1 shows the general characteristics of the older adults included. There were more women with probable sarcopenia than non-sarcopenic ones (81.6% vs. 41.7%; *p* = 0.026). Thus, there were statistically significant differences in weight (*p* < 0.001), height (*p* < 0.001), and BMI (*p* = 0.032) between older adults with probable sarcopenia and non-sarcopenic individuals.

Regarding the muscle strength assessment (Table 2), the HGS was statistically significantly higher in non-sarcopenic older adults (*p* < 0.001) compared to those with probable sarcopenia, and the significance was maintained by sex (men *p* = 0.002; women *p* < 0.001). No statistically significant differences were observed in the gait speed between probable sarcopenic and non-sarcopenic individuals (*p* > 0.05).

### 3.1. Muscle Mass Assessed by BIA

Focusing on muscle mass quantity assessed by the BIA, the SM (*p* < 0.001), SMI (*p* < 0.001), ASM (kg) (*p* < 0.001), and ASMI (*p* = 0.002) were statistically significantly higher in non-sarcopenic older adults. In addition, the significance was maintained only in women (SM (*p* < 0.001), SMI (*p* = 0.003), ASM (kg) (*p* = 0.001), and ASMI (*p* = 0.020)) when segregated by sex. Regarding the muscle mass quality, the phase angle was statistically significantly higher in non-sarcopenic older adults compared to those with probable sarcopenia (*p* = 0.003). Also, the significance was maintained in only in men (*p* = 0.041) when segregated by sex (Table 2).

### 3.2. Muscle Mass Assessed by Isokinetic Parameters

Regarding the leg isokinetic assessment, the PT, TW, and MP from extension (PT *p* < 0.001; TW *p* < 0.001; MP *p* = 0.002) and flexion (PT *p* = 0.007; TW *p* = 0.007; MP *p* = 0.011) at 180° s^−1^ were statistically significantly higher in non-sarcopenic older adults compared to those with probable sarcopenia (*p* < 0.05). Moreover, the PT from extension was statistically significantly higher in non-sarcopenic men (*p* = 0.028) and women (*p* = 0.024) when segregated by sex, and MP from extension only in men (*p* = 0.042). The significance was maintained after adjusting according to sex in all isokinetic parameters (*p* < 0.001) (Table 3).

### 3.3. Muscle Mass Assessed by Ultrasound

Regarding the ultrasound assessment, the RF-P at 30% (*p* = 0.014) and the FA-MT (*p* = 0.001) were statistically significantly higher in non-sarcopenic older adults in comparison to those with probable sarcopenia. Focusing on the SFT, non-sarcopenic older adults had statistically significantly lower SFT at 30% compared to non-sarcopenic individuals (*p* = 0.004). The significance of the RF-P at 30%, FA-MT and the SFT at 30% was maintained after adjusting according to sex (*p* < 0.001).

In addition, no statistically significant differences were observed for the other ultrasound parameters among older adults comparing sarcopenia conditions. However, after adjusting according to sex, the QF-MT at 30%, RF-MT at 30% and 50%, RF-W at 30%, RF-CSA at 30%, and QF-MT at 50% were statistically significantly higher in non-sarcopenic older adults compared to those with probable sarcopenia (*p* < 0.001). Also, SFT at 50% was statistically significantly lower in non-sarcopenic older adults compared to older adults with probable sarcopenia after adjusting according to sex (*p* < 0.001). No statistically significant differences were observed in the upper-leg pennation angle (*p* > 0.05) (Table 3).

### 3.4. ROC Analysis and Cut-Off Points to Confirm the Sarcopenia Diagnosis

Regarding muscle mass parameters measured by the BIA, the ROC analysis revealed that the SM, SMI, and ASM (kg) had an excellent diagnostic accuracy of sarcopenia in women with probable sarcopenia (AUC ≥ 0.9; *p* < 0.001). And, also, the ASMI had a good diagnostic accuracy of sarcopenia in women with probable sarcopenia (AUC = 0.826; *p* < 0.001) (Appendix A Figure A2 and Table 4). According to Youden’s index, the best cut-off points for these parameters to confirm the sarcopenia diagnosis in women with probable sarcopenia were 24.35 kg for SM, 9.88 kg/m^2^ for SMI, 18.75 kg for ASM, and 6.85 kg/m^2^ for ASMI. No statistically significant ROC analysis was obtained for men (*p* > 0.05). Moreover, the ROC analysis showed that the phase angle had a good diagnostic accuracy of sarcopenia in men (AUC = 0.833; *p* = 0.003) and women (AUC = 0.835; *p* < 0.001) with probable sarcopenia. The best cut-off points for the phase angle based on Youden’s index to confirm sarcopenia diagnosis were 5.1° in men and 4.95° in women with probable sarcopenia (Figure 2a,b and Table 4).

Regarding the isokinetic parameters in the extension, the PT in men and the TW and MP in both men and women had a good diagnostic accuracy of sarcopenia (AUC = 0.80–0.89; *p* < 0.05), and the PT in women had an excellent diagnostic accuracy of sarcopenia (AUC ≥ 0.9; *p* < 0.05) in older adults with probable sarcopenia. According to Youden’s index, the best cut-off points to confirm sarcopenia diagnosis were 66.75 N-m for PT, 64.0 J for TW, and 87.8 W for MP in men with probable sarcopenia. The cut-off points in women with probable sarcopenia were 48.35 N-m for PT, 54.70 J for TW, and 48.95 W for MP (Figure 3 and Table 4). No statistically significant ROC analysis was obtained for other isokinetic parameters in flexion (*p* > 0.05) (Appendix B Table A1). 

Focusing on ROC analysis of ultrasound parameters, the upper-leg pennation angle in men and the FA-MT in both men and women had a fair diagnostic accuracy of sarcopenia in older adults with probable sarcopenia (AUC = 0.70–0.79; *p* < 0.05). The best cut-off points for these parameters based on Youden’s index to confirm sarcopenia diagnosis was 10.11° for the upper-leg pennation angle in men, 1.41 cm for the FA-MT in men, and 0.94 cm for the FA-MT in women with probable sarcopenia (Figure 2c–e and Table 4). No statistically significant ROC analysis was obtained for other ultrasound parameters (*p* > 0.05) (Appendix B Table A1).

## 4. Discussion

The present cross-sectional exploratory analysis study adds evidence about the diagnostic accuracy of sarcopenia by muscle mass quantity and quality parameters, using the BIA, isokinetic, and ultrasound tools, in community-dwelling older adults with probable sarcopenia. Also, some cut-off points for these parameters are suggested to be more representative of the population of older adults with probable sarcopenia as a starting point for an improvement in the confirmation of the diagnosis of sarcopenia. In particular, the parameters of phase angle by BIA, isokinetic PT, TW, and MP in leg extension, as well as FA-MT by ultrasound, seem to have great diagnostic accuracy of sarcopenia based on the AUC from the ROC analysis.

The EWGSOP2 consensus proposes the evaluation of the quality of muscle mass combined with the quantity of the muscle mass as an important aspect to confirm the sarcopenia diagnosis, but it does not suggest any validated cut-off point [1]. Thus, given this gap in the literature, it could be possible that some older adults who are categorized as probable sarcopenic with no quantity of muscle mass alteration already have an alteration in the quality of the muscle mass and could be underdiagnosed in regard to sarcopenia. Therefore, if new cut-off points and ways of evaluating muscle mass quality are proposed, these people could be diagnosed as sarcopenic. 

Regarding muscle mass quality assessed by BIA, the present study adds evidence that the phase angle obtained from BIA has a great diagnostic accuracy of sarcopenia in both sexes with probable sarcopenia. The scientific literature considered that the phase angle is a good parameter to detect early alterations in muscle mass quality because it decreases earlier than other body-composition parameters, such as muscle mass quantity [20]. Furthermore, we also proposed a possible cut-off point for the phase angle (5.1° in men and 4.95° in women) as a useful parameter to confirm the sarcopenia diagnosis based on muscle mass quality by BIA in older adults with probable sarcopenia. Although there are no validated cut-off points for the phase angle, our proposed cut-off points are in concordance with other studies that proposed a cut-off point that ranges from 4.05° to 6.12° in men and from 3.55° to 5.74° in women [21,22].

Regarding muscle mass quantity parameters measured by BIA, the SM, SMI, ASM, and ASMI have a great diagnostic accuracy of sarcopenia in women with probable sarcopenia, and a proposal of cut-off points was provided (SM, 24.35 kg; SMI, 9.88 kg/m^2^; ASM, 18.75 kg; and ASMI, 6.85 kg/m^2^). However, the cut-off points of muscle mass quantity parameters assessed via the BIA proposed in the present study are higher than the cut-off points of EWGSOP1 and EWGSOP2 [1,2], so it could be an explanation for the difficulty in confirming the diagnosis of sarcopenia. 

Focusing on isokinetic assessment is a versatile technique that allows us to assess the muscle strength of lower limbs, as well as to evaluate muscle function as a parameter of the quality of muscle mass [6,9]. The evidence supports that the PT and MP of the knees were useful parameters to assess the muscle strength of the lower limbs, as well as physical performance, based on balance and mobility [23]. The results of the present study established that the PT, TW, and MP in extension have a great diagnostic accuracy of sarcopenia in individuals with probable sarcopenia. Despite the fact that there is no validated cut-off point at an angular velocity of 180° s^−1^ in extension and flexion in the scientific literature, the present study proposes some possible cut-off points to confirm the diagnoses of sarcopenia for all isokinetic parameters in extension both in men and women with probable sarcopenia (PT, 66.75 N-m in men, and 48.35 N-m in women; TW, 64.0 J in men, and 54.70 J in women; and MP, 87.8 (W) in men, and 48.95 W in women). 

Ultrasound is a technique that brings information about the muscle structure, using, for example, the pennation angle, or the muscle composition, such as intramuscular adipose tissue infiltration or ecogenicity [6,10]. Moreover, according to two systematic reviews, ultrasound is a technique with high methodological heterogeneity [12,13]. A standardized protocol for ultrasound that focuses on the patient’s position, the ultrasound probe used, and the muscular parameters assessed is necessary [12,13].

The present study determined that the upper leg pennation angle muscle mass quality parameter assessed by ultrasound has a great diagnostic accuracy of sarcopenia only in men with probable sarcopenia and provided a possible cut-off point (10.11°) to confirm sarcopenia diagnosis. Additionally, the scientific evidence also proposed the ultrasound measurement of vastus lateralis as a useful measure to assess the aging effects on muscle, in particular, the ultrasound sarcopenia index, which is the ratio between the muscle thickness of the vastus lateralis and the fascicles length [24,25].

Also, the FA-MT has a great diagnostic accuracy of sarcopenia in older adults with probable sarcopenia, and possible cut-off points for sarcopenia diagnosis based on the upper limbs as FA-MT was provided for both sexes (1.41 cm in men; 0.94 cm in women). However, a cut-off point could not be established for the other upper-leg ultrasound parameters. Although there is scarce evidence about upper-limb ultrasound, a systematic review and meta-analysis determined that FA-CSA has a moderate diagnostic accuracy of sarcopenia [26,27]. In addition, ultrasound assessment of FA is easier to perform in clinical practice compared with upper-limb ultrasound [18].

Figure 4 shows new perspectives for low muscle mass quantity and quality assessments to increase the accuracy of sarcopenia diagnosis in community-dwelling older adults. For the community-dwelling older adults from the present study, the muscle mass quantity parameters assessed by BIA are unaltered, probably because the cut-off points of the EWGSOP1 and EWGSOP2 consensus are not representative of this population. For this reason, the present study provides new parameters to assess the quantity and quality of the muscle mass and also proposes new possible cut-off points, solving the current gap in the literature. Thus, the parameters identified as promising to improve the diagnostic accuracy of sarcopenia are the phase angle by BIA, isokinetic peak torque, total work, and mean power in leg extension, and forearm muscle thickness by ultrasound. With the evaluation of these parameters, the complaints of the individuals in the present study on the functional domain will be understood, such as climbing stairs, representing a useful translation strategy into clinical practice. 

The present study has some strengths. The older adults included were individuals with probable sarcopenia, and this allowed us to find cut-off points for other parameters of sarcopenia that can be applied to individuals in an early stage of sarcopenia. In addition, this study may be the basis for improving the current consensus on sarcopenia in older adults. Moreover, the target population was community-dwelling older adults. This includes the general population of older adults but, at the same time, makes the diagnosis of sarcopenia more difficult compared to institutionalized older adults.

However, this study has some limitations too. First, the sample size is small, and the sample of non-sarcopenic older adults was smaller than the rest of those with probable sarcopenia. Second, there was an imbalance between men and women. This was because it was much more difficult to include men with probable sarcopenia, as they were above the sarcopenia parameter cut-off points for inclusion by the EWGSOP1 consensus. Third, there is a low effect size due to the small sample size. It would be interesting to improve the analysis with a larger sample.

## 5. Conclusions

In conclusion, the muscle mass parameters of quantity and quality as the phase angle by BIA, isokinetic peak torque, total work, and mean power in leg extension, as well as the forearm muscle thickness by ultrasound, had a great diagnostic accuracy of sarcopenia, and their suggested cut-off points could be useful to confirm the diagnosis of sarcopenia in both sexes in probable sarcopenic individuals. However, more research is needed with a higher sample size to confirm this. These results could open a novel strategy to increase the accuracy of sarcopenia diagnosis in community-dwelling older adults. Further research is needed to determine which parameter is altered first and elucidate the role of the muscle mass quality in improving the sarcopenia diagnosis in older adults. 

## Figures and Tables

**Figure 1 nutrients-16-01496-f001:**
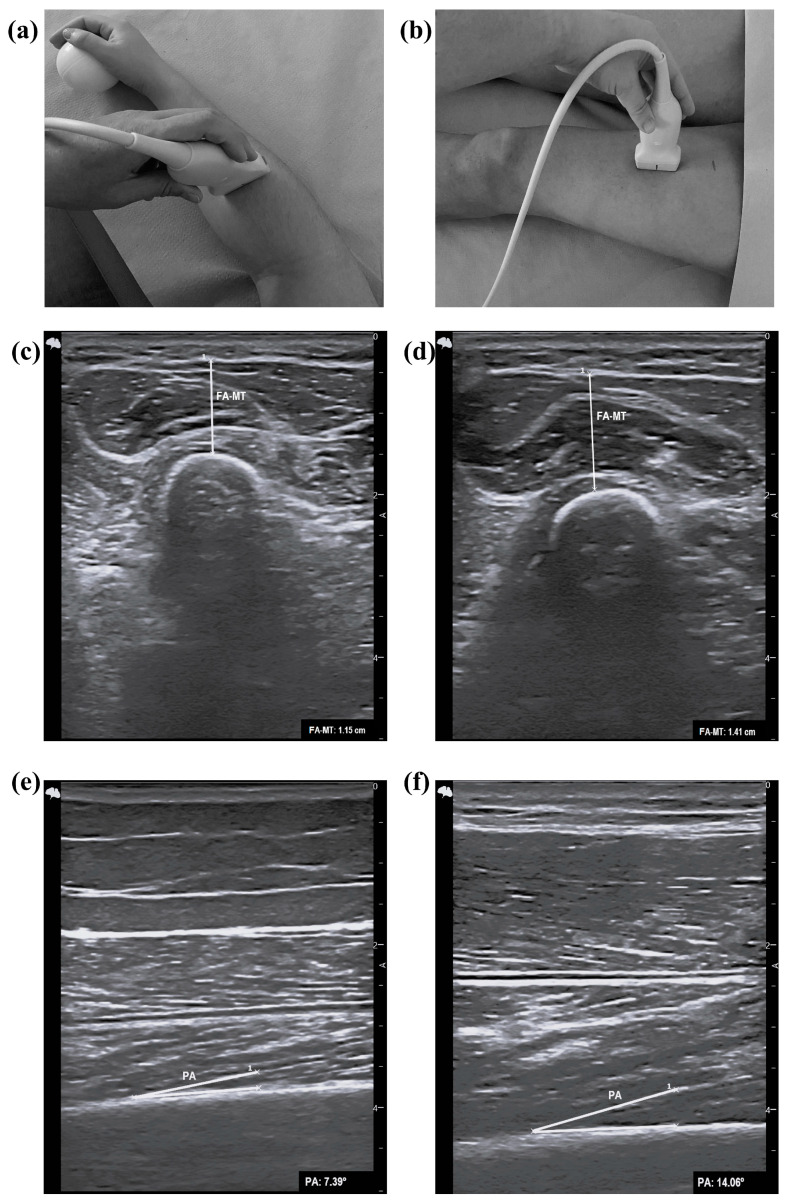
Measurement of FA and pennation angle ultrasound parameters. (**a**) Transducer position at 30% proximal between the styloid process of the radius and the insertion of the biceps brachii muscle into the radial tuberosity. (**b**) Transducer position 30–50% between the anterior super iliac spine and the proximal end of the patella with transduce parallel to the long axis. (**c**) FA-MT of probable sarcopenic older adults. (**d**) FA-MT of non-sarcopenic older adults. (**e**) Upper-leg pennation angle of probable sarcopenic older adults. (**f**) Upper-leg pennation angle of non-sarcopenic older adults. FA-MT, forearm muscle thickness; PA, pennation angle.

**Figure 2 nutrients-16-01496-f002:**
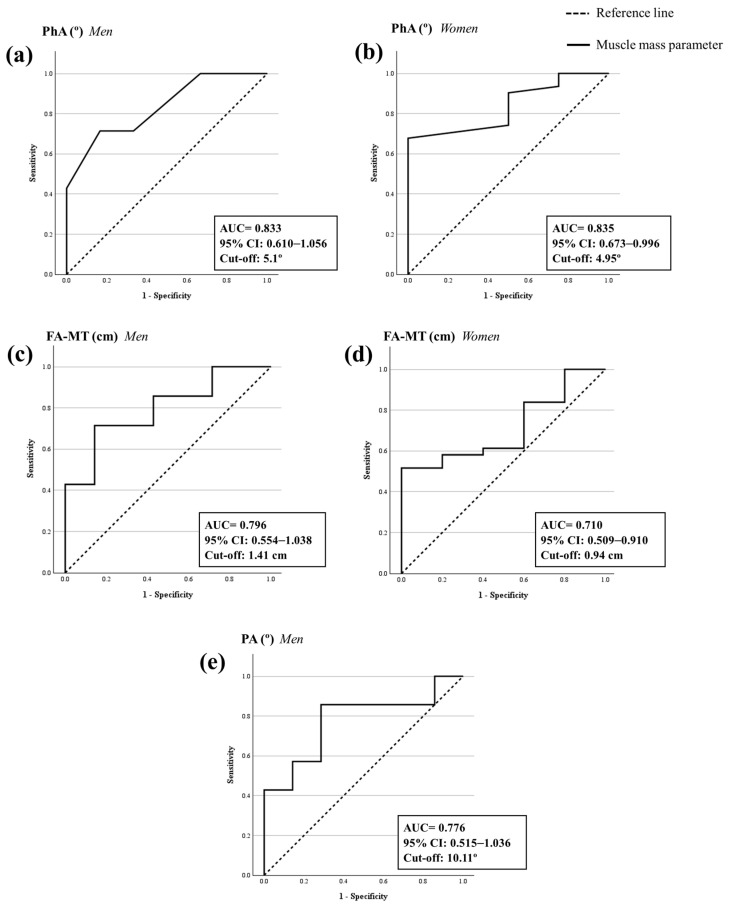
ROC analysis for the diagnostic accuracy of sarcopenia of the phase angle by BIA and ultrasound parameters. (**a**) Phase angle (PhA) by BIA in men. (**b**) Phase angle (PhA) by BIA in women. (**c**) Forearm muscle thickness (FA-MT) in men. (**d**) Forearm muscle thickness (FA-MT) in women. (**e**) Pennation angle (PA) in men. BIA: bioelectrical impedance analysis. ROC analysis is significant when the area under the curve (AUC) is >0.5 and the lower 95% confidence interval (CI) value is >0.5. A *p*-value < 0.05 is statistically significant.

**Figure 3 nutrients-16-01496-f003:**
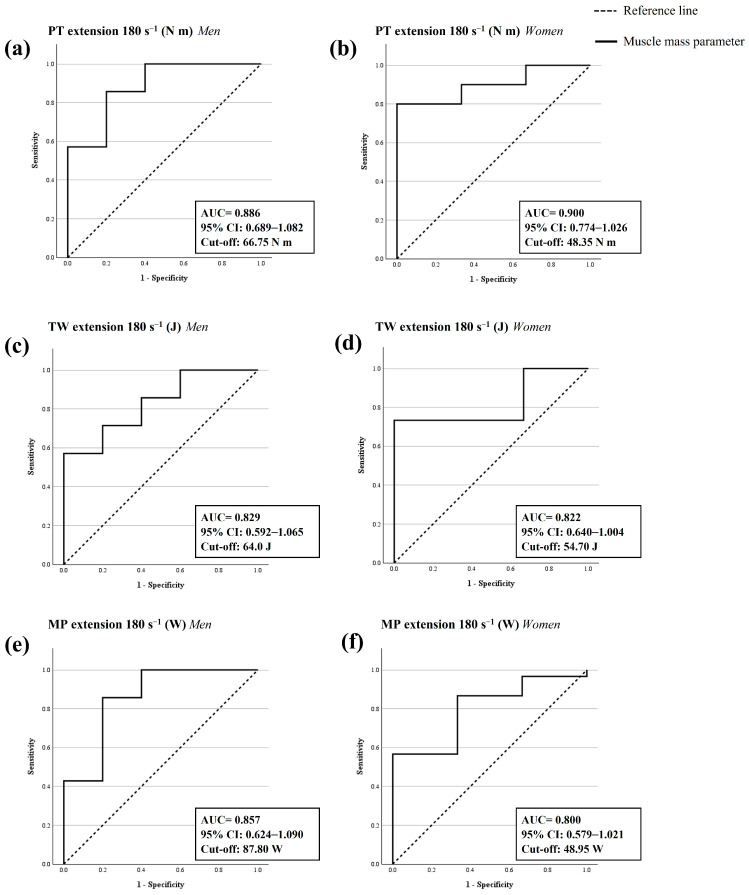
ROC analysis for the diagnostic accuracy of sarcopenia of isokinetic parameters. (**a**) Peak torque (PT) in extension at 180° s^−1^ in men. (**b**) Peak torque (PT) in extension at 180° s^−1^ in women. (**c**) Total work (TW) in extension at 180° s^−1^ in men. (**d**) Total work (TW) in extension at 180° s^−1^ in women. (**e**) Mean power (MP) in extension at 180° s^−1^ in men. (**f**) Mean power (MP) in extension at 180° s^−1^ in women. ROC analysis is significant when the area under the curve (AUC) is >0.5 and the lower 95% confidence interval (CI) value is >0.5. A *p*-value < 0.05 is statistically significant.

**Figure 4 nutrients-16-01496-f004:**
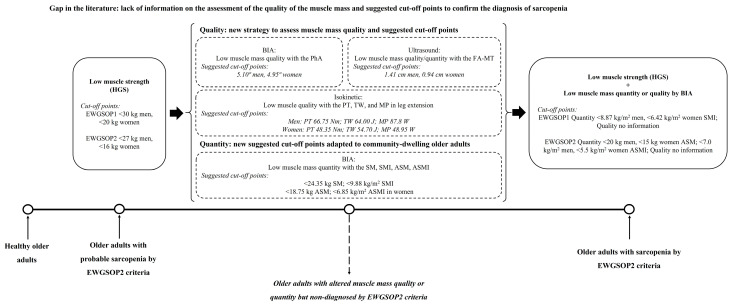
New perspectives for low muscle mass quantity and quality assessment to increase the accuracy of sarcopenia diagnosis in community-dwelling older adults. EWGSOP, European Working Group on Sarcopenia in Older People; HGS, handgrip strength; BIA, bioelectrical impedance analysis; PhA, phase angle; FA-MT, forearm muscle thickness; PT, peak torque; TW, total work; MP, mean power; SM, skeletal muscle mass; SMI, skeletal muscle mass index; ASM, appendicular skeletal muscle mass; ASMI, appendicular skeletal muscle mass index.

**Table 1 nutrients-16-01496-t001:** General characteristics of the included older adults.

	Probable Sarcopenia (*n* = 38)	Non-Sarcopenic (*n* = 12)	*p*-Value
Age (years)	69.6 ± 4.1	67.6 ± 4.5	0.151 ^a^
Sex *n* (%)			
Men	7 (18.4)	7 (58.3)	0.026 ^a^
Women	31 (81.6)	5 (41.7)	
Weight (kg)	67.4 ± 13.1	82.6 ± 9.7	<0.001 ^b^
Height (m)	1.56 (0.08)	1.69 (0.08)	<0.001 ^b^
BMI (kg/m^2^)	27.27 ± 4.21	29.46 ± 2.89	0.032 ^b^

BMI, body mass index. Values are mean ± standard deviation (mean ± SD) or median and interquartile range, unless indicated. ^a^ *t*-test. ^b^ Mann–Whitney U test. A *p*-value < 0.05 is statistically significant.

**Table 2 nutrients-16-01496-t002:** Sarcopenia parameters of older adults with probable sarcopenia and non-sarcopenic.

		Probable Sarcopenia (*n* = 38) ^a^	Non-Sarcopenic (*n* = 12) ^b^	*p*-Value ^c^	Effect Size
Muscle strength and physical performance					
HGS (kg)	All	17.68 (3.82)	35.38 (13.04)	<0.001	0.015
	Men	26.90 (10.10)	40.15 (10.80)	0.002	
	Women	17.15 (3.75)	28.90 (5.75)	<0.001	
GS (m/s)	All	1.03 ± 0.18	1.12 ± 0.16	0.266	0.393
	Men	1.10 ± 0.08	1.20 ± 0.15	0.142	
	Women	1.02 ± 0.20	1.00 ± 0.10	0.631	
Quantity of muscle mass					
SM (kg)	All	22.95 (3.76)	33.50 (7.60)	<0.001	0.122
	Men	33.50 (5.20)	34.95 (2.85)	0.224	
	Women	22.30 (2.40)	27.40 (2.10)	<0.001	
SMI (kg/m^2^)	All	9.64 (0.88)	11.27 (1.75)	<0.001	0.165
	Men	12.05 (2.16)	11.86 (1.09)	0.830	
	Women	9.49 (0.78)	10.22 (0.70)	0.003	
ASM (kg)	All	16.25 (2.43)	24.80 (7.20)	<0.001	0.132
	Men	24.80 (5.00)	26.15 (2.80)	0.283	
	Women	15.80 (1.80)	19.10 (1.25)	0.001	
ASM (%)	All	26.25 (4.58)	29.20 (4.90)	0.098	0.335
	Men	29.70 (2.20)	29.90 (1.45)	0.475	
	Women	25.00 (3.40)	25.10 (2.10)	0.982	
ASMI (kg/m^2^)	All	6.80 (0.63)	8.40 (1.90)	0.002	0.189
	Men	8.70 (2.00)	8.95 (1.00)	0.774	
	Women	6.70 (0.50)	7.10 (0.60)	0.020	
Quality of muscle mass					
PhA (°)	All	4.76 ± 0.51	5.38 ± 0.33	0.003	0.208
	Men	4.86 ± 0.40	5.38 ± 0.31	0.041	
	Women	4.74 ± 0.54	5.38 ± 0.41	0.124	

HGS, handgrip strength; GS, gait speed; SM, skeletal muscle mass; SMI, skeletal muscle mass index; ASM, appendicular skeletal muscle mass; ASMI, appendicular skeletal muscle mass index; PhA, phase angle. Values are mean ± standard deviation (mean ± SD) or median and interquartile range. ^a^ Men (*n* = 7) and women (*n* = 31). ^b^ Men (*n* = 7) and women (*n* = 5), excluding SM, SMI, ASM, and ASMI (*n* = 6 men) and PhA (*n* = 4 women). ^c^ Mann–Whitney U test. A *p*-value < 0.05 is statistically significant.

**Table 3 nutrients-16-01496-t003:** Ultrasound and isokinetic parameters of older adults with probable sarcopenia and non-sarcopenic.

		Probable Sarcopenia (*n* = 38) ^a^	Non-Sarcopenic (*n* = 12) ^b^	*p*-Value ^c^	*p*-Value Adjusted ^d^	Effect Size
Ultrasound						
Upper leg at 30%						
SFT (cm)	All	1.28 ± 0.57	0.80 ± 0.39	0.004	<0.001	0.221
	Men	0.75 ± 0.58	0.54 ± 0.18	0.654		
	Women	1.41 ± 0.50	1.16 ± 0.29	0.203		
QF-MT (cm)	All	2.31 (0.89)	2.81 (1.35)	0.058	<0.001	0.316
	Men	2.97 (0.30)	3.20 (1.20)	0.565		
	Women	2.14 (0.74)	2.20 (0.38)	0.572		
RF-MT (cm)	All	1.32 (0.45)	1.33 (0.82)	0.205	<0.001	0.377
	Men	1.79 (0.75)	1.90 (0.75)	0.848		
	Women	1.15 (0.43)	1.10 (0.24)	0.621		
RF-W (cm)	All	3.89 ± 0.38	4.05 ± 0.22	0.109	<0.001	0.345
	Men	4.37 ± 0.22	4.15 ± 0.17	0.064		
	Women	3.77 ± 0.32	3.91 ± 0.22	0.346		
RF-CSA (cm^2^)	All	4.06 (1.60)	4.67 (3.10)	0.085	<0.001	0.333
	Men	7.02 (3.47)	6.61 (2.33)	0.749		
	Women	3.62 (1.38)	3.90 (0.91)	0.850		
RF-P (cm)	All	8.97 (0.92)	9.75 (1.69)	0.014	<0.001	0.261
	Men	11.12 (1.91)	10.60 (0.85)	0.655		
	Women	8.82 (0.73)	9.28 (0.76)	0.120		
Upper leg at 50%						
SFT (cm)	All	1.45 ± 0.59	1.08 ± 0.47	0.061	<0.001	0.319
	Men	0.92 ± 0.71	0.76 ± 0.17	0.337		
	Women	1.57 ± 0.49	1.52 ± 0.36	0.981		
QF-MT (cm)	All	3.29 ± 0.71	3.80 ± 1.14	0.264	<0.001	0.392
	Men	4.04 ± 0.71	4.49 ± 0.97	0.406		
	Women	3.12 ± 0.59	2.82 ± 0.37	0.220		
RF-MT (cm)	All	1.64 (0.33)	1.93 (0.95)	0.114	<0.001	0.347
	Men	2.25 (0.89)	2.32 (0.55)	0.848		
	Women	1.62 (0.28)	1.53 (0.47)	0.423		
Upper leg						
PA (°)	All	9.63 ± 3.32	10.56 ± 2.26	0.317	0.597	0.403
	Men	8.14 ± 3.20	11.40 ± 2.52	0.085		
	Women	9.98 ± 3.30	9.38 ± 1.21	0.741		
Forearm						
FA-MT (cm)	All	1.02 ± 0.33	1.48 ± 0.40	0.001	<0.001	0.184
	Men	1.36 ± 0.30	1.69 ± 0.33	0.064		
	Women	0.94 ± 0.29	1.17 ± 0.27	0.137		
Isokinetic						
Leg extension at 180° s^−1^						
PT (N-m)	All	43.40 (16.55)	65.55 (34.65)	<0.001	<0.001	0.100
	Men	61.80 (12.80)	87.20 (25.30)	0.028		
	Women	41.90 (13.65)	52.00 (-)	0.024		
TW (J)	All	48.75 ± 17.84	76.56 ± 18.52	<0.001	<0.001	0.125
	Men	63.76 ± 19.99	85.34 ± 16.54	0.062		
	Women	45.25 ± 15.67	61.93 ± 11.84	0.069		
MP (W)	All	49.26 ± 19.92	87.25 ± 31.33	0.002	<0.001	0.142
	Men	65.69 ± 23.18	102.3 ± 29.92	0.042		
	Women	45.43 ± 17.35	62.17 ± 11.61	0.091		
Leg flexion at 180° s^−1^						
PT (N-m)	All	27.20 (11.20)	38.40 (23.08)	0.007	<0.001	0.191
	Men	37.30 (18.70)	51.90 (20.35)	0.465		
	Women	26.25 (8.95)	30.30 (-)	0.222		
TW (J)	All	22.53 ± 12.32	37.99 ± 14.68	0.007	<0.001	0.191
	Men	34.27 ± 16.85	45.70 ± 12.68	0.291		
	Women	19.79 ± 9.41	25.13 ± 6.01	0.234		
MP (W)	All	20.60 (20.90)	41.40 (36.25)	0.011	<0.001	0.209
	Men	35.90 (15.50)	57.70 (39.55)	0.167		
	Women	18.40 (18.03)	24.80 (-)	0.316		

QF, quadriceps femoris; PA, pennation angle; SFT, subcutaneous fat thickness; QF-MT, quadriceps femoris muscle thickness; RF-MT, rectus femoris muscle thickness; RF-W, rectus femoris width; RF-CSA, rectus femoris cross-sectional area; RF-P, rectus femoris perimeter; FA-MT, forearm muscle thickness; PT, peak torque; TW, total work; MP, mean power. Values are mean ± standard deviation (mean ± SD) or median and interquartile range. ^a^ Men (*n* = 7) and women (*n* = 31), excluding upper-leg parameters and for isokinetic parameters (women *n* = 30). ^b^ Men (*n* = 7) and women (*n* = 5), excluding isokinetic parameters (men *n* = 5 and women *n* = 3). ^c^ Mann–Whitney U test. ^d^ Univariate analysis adjusted by sex. A *p*-value < 0.05 is statistically significant.

**Table 4 nutrients-16-01496-t004:** ROC analysis for the diagnostic accuracy of sarcopenia by quantity and quality muscle mass parameters using the BIA, ultrasound, and isokinetic tools and cut-off points for the sarcopenia diagnosis.

	AUC	95% CI	*p*-Value	Sensitivity	Specificity	Youden’s Index	Cut-Off
BIA							
SM (kg)							
Men	0.702	0.401–1.004	0.188	-	-	-	-
Women	0.971	0.910–1.032	<0.001	0.839	0.000	0.839	24.35
SMI (kg/m^2^)							
Men	0.464	0.135–0.794	0.832	-	-	-	-
Women	0.913	0.805–1.021	<0.001	0.774	0.000	0.774	9.88
ASM (kg)							
Men	0.679	0.370–0.987	0.256	-	-	-	-
Women	0.958	0.875–1.041	<0.001	1.000	0.200	0.800	18.75
ASM (%)							
Men	0.619	0.298–0.940	0.468	-	-	-	-
Women	0.497	0.301–0.693	0.974	-	-	-	-
ASMI (kg/m^2^)							
Men	0.548	0.219–0.876	0.776	-	-	-	-
Women	0.826	0.650–1.002	<0.001	0.710	0.200	0.510	6.85
PhA (°)							
Men	0.833	0.610–1.056	0.003	0.714	0.167	0.548	5.10
Women	0.835	0.673–0.996	<0.001	0.677	0.000	0.677	4.95
Ultrasound							
Upper leg PA (°)							
Men	0.776	0.515–1.036	0.038	0.857	0.286	0.571	10.11
Women	0.453	0.259–0.648	0.639	-	-	-	-
FA-MT (cm)							
Men	0.796	0.554–1.038	0.017	0.714	0.143	0.571	1.41
Women	0.710	0.509–0.910	0.041	0.516	0.000	0.516	0.94
Isokinetic							
PT in leg extension at 180° s^−1^ (N-m)							
Men	0.886	0.689–1.082	<0.001	0.857	0.200	0.657	66.75
Women	0.900	0.774–1.026	<0.001	0.800	0.000	0.800	48.35
TW in leg extension at 180° s^−1^ (J)							
Men	0.829	0.592–1.065	0.007	0.571	0.000	0.571	64.00
Women	0.822	0.640–1.004	0.001	0.733	0.000	0.733	54.70
MP in leg extension at 180° s^−1^ (W)							
Men	0.857	0.624–1.090	0.003	0.857	0.200	0.657	87.8
Women	0.800	0.579–1.021	0.008	0.567	0.000	0.567	48.95

AUC, area under the curve; CI, confidence interval; BIA, bioelectrical impedance analysis; SM, skeletal muscle mass; SMI, skeletal muscle mass index; ASM, appendicular skeletal muscle mass; ASMI, appendicular skeletal muscle mass index; PhA, phase angle; PA, pennation angle; FA-MT, forearm muscle thickness; PT, peak torque; TW, total work; MP, mean power. ROC analysis is significant when AUC is >0.5 and the lower 95% CI value is >0.5. A *p*-value < 0.05 is statistically significant.

## Data Availability

The data presented in this study are available upon request from the corresponding author due to privacy reasons.

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
