# Peer review of "New Perspectives for Low Muscle Mass Quantity/Quality Assessment in Probable Sarcopenic Older Adults: An Exploratory Analysis Study"

_nutrients, 2024, doi:10.3390/nu16101496_

Round 1
Reviewer 1 Report
Comments and Suggestions for Authors
This manuscript describes the results of a prospective study that aims to provide additional information about different parameters including neuromuscular and ultrasound measurements, in individuals with probable sarcopenia. Despite the topic being of interest and the authors can be appreciated for the large number of measurements they have taken, the main limits of this manuscript are the small sample size, the unequal distribution in the two groups, and the presence of male and female subjects that, in this small sample size, can strongly influence the findings (and I think that the proposed statistical model to "correct" the bias due to sex may not be sufficient). Therefore, the manuscript must be reported as an "exploratory study", and in my opinion, it is not possible to suggest cut-off values and diagnostic accuracy with such limitations. To provide an opportunity for publication, I suggest being more careful with the findings' interpretation and maybe sticking to the comparison of the different assessed parameters between the two groups, otherwise, the risk for misinterpretation is really high.
I have some comments that in my modest opinion should be considered
Abstract:
- In the results, please report within the included sample how many participants were grouped in the two categories according to sarcopenia status and their main demographics (age and sex)
- I recommend providing full p values (2 or 3 decimals) instead of reporting < 0.05; this is usually discouraged by statistical guidelines.
- Here in the abstract and throughout the whole manuscript, I suggest to substitute gender with sex, as the second refers to the biological sex and it is recommended to discriminate the use of gender and sex as not equal.
Statistical analysis:
In general, this section is too long and clumsy. Be careful to not replicate what is already present in tables or figures in the text.
- Since the different sample sizes in the two groups (that are not very large themselves), I think that non-parametric testing should be performed to provide more robust statistical support.
- The authors performed different US measurements on the same muscle; as such, I think that this leads to a possible repeated measures error and a correction should be applied.
- Reporting the effect size of the performed statistical analysis should be recommended, especially due to the limited sample size.
Results:
- Please, be consistent: men and women, or males and females (Table 1)
- Age can be reported with only 1 decimal; the same for body weight. In general, the number of decimals should be based on the precision of the instrument.
- In the notes below the table, it is usually recommended to report also the statistical test used to provide the significance.
- Line 249, please report full p-value, not p< 0.05
Discussion:
- Overall, I think that the authors should be extremely careful with the interpretation of their findings. As they also recognize in the short limitations section, the small sample size, with not equal proportions of sarcopenic and non-sarcopenic individuals, and the unbalanced presence of males and females, does not encourage the performance of statistical analysis that recommends cut-off values. Unfortunately, this is a big bias that must be mentioned, and the whole study needs to be somehow "downgraded" to an exploratory analysis (which should be well stated in the title of the manuscript, abstract and main text). Usually, this kind of studies recruit far more than 100 people to reduce interindividual variability.
- Regarding the US protocol, the authors should be aware that vastus lateralis has been often recommended for the thigh muscles, as it better reflects aging or disuse alterations; in addition, the Ultrasound Sarcopenia Index integrates pennation angle and muscle thickness to limit the bias due to different anthropometrics and sex (Narici et al., 2021; Di Lenarda et al., 2024). I suggest discussing these limitations too according to the literature.
Author Response
Nutrients Manuscript: nutrients-2993433
Title: New perspectives for low muscle mass quantity/quality assessment in probable sarcopenic older adults: A cross-sectional study
Authors: Maria Besora-Moreno, Elisabet Llauradó*, Claudia Jiménez-ten Hoevel, Cristina Sepúlveda, Judit Queral, Glòria Bernal, Laura Pérez-Merino, Blanca Alabadi, Yolanda Ortega, Rosa Maria Valls, Rosa Solà*, and Anna Pedret
Dear Editor,
Thank you for your letter and the reviewers’ comments on our manuscript entitled “New perspectives for low muscle mass quantity/quality assessment in probable sarcopenic older adults: A cross-sectional study”.
We appreciate the time and effort that you and the reviewers have dedicated to providing your valuable feedback on our manuscript and we are grateful to the reviewers for their insightful comments on our paper.
Finally, we have been able to address all the issues suggested by reviewers as well as to incorporate into the original manuscript all the proposed changes. Accordingly, we have added the appropriate clarifications in the revised manuscript, which has allowed us to improve the manuscript content. We have highlighted the changes suggested by the reviewers in yellow color within the revised manuscript.
Also, according to the reviewer’s suggestion we changed the title of the manuscript as “New perspectives for low muscle mass quantity/quality assessment in probable sarcopenic older adults: An exploratory analysis study”. Also, we have included a new author that we mistakenly forgot when we submitted the manuscript.
We hope that the revised manuscript meets the quality standard required for publication in Nutrients.
Sincerely,
Elisabet Llauradó and Rosa Solà
Functional Nutrition, Oxidation, and Cardiovascular Diseases Group (NFOC-Salut)
Facultat de Medicina i Ciències de la Salut
Universitat Rovira i Virgili
43201 Reus
Spain
Tel.: (0034) 977758920
E-mail: elisabet.llaurado@urv.cat; rosa.sola@urv.cat
REVIEWER 1
Comments and Suggestions for Authors
This manuscript describes the results of a prospective study that aims to provide additional information about different parameters including neuromuscular and ultrasound measurements, in individuals with probable sarcopenia. Despite the topic being of interest and the authors can be appreciated for the large number of measurements they have taken, the main limits of this manuscript are the small sample size, the unequal distribution in the two groups, and the presence of male and female subjects that, in this small sample size, can strongly influence the findings (and I think that the proposed statistical model to "correct" the bias due to sex may not be sufficient). Therefore, the manuscript must be reported as an "exploratory study", and in my opinion, it is not possible to suggest cut-off values and diagnostic accuracy with such limitations. To provide an opportunity for publication, I suggest being more careful with the findings' interpretation and maybe sticking to the comparison of the different assessed parameters between the two groups, otherwise, the risk for misinterpretation is really high.
I have some comments that in my modest opinion should be considered
Abstract:
Question 1. In the results, please report within the included sample how many participants were grouped in the two categories according to sarcopenia status and their main demographics (age and sex)
Response 1. As the reviewer suggests we have included information about sarcopenia status, age, and sex in the abstract section (page 1, lines 33-34): “38 with probable sarcopenia (69.63±4.14 years; 7 men and 31 women) and 12 non-sarcopenic (67.58±4.54 years; 7 men and 5 women).”.
Question 2. I recommend providing full p values (2 or 3 decimals) instead of reporting < 0.05; this is usually discouraged by statistical guidelines.
Response 2. We have improved the abstract section including all p-values (pages 1, lines 34-39): “The phase angle (cut-off: 5.10º men, p=0.003; 4.95º women, p<0.001), peak torque (cut-off: 66.75 Newtons-meters (N-m) men, p<0.001; 48.35 N-m women, p<0.001), total work (cut-off: 64.00 Joules (J) men, p=0.007; 54.70 J women, p=0.001), and mean power (cut-off: 87.8 Watts (W) men, p=0.003; 48.95 W women, p=0.008) in leg extension, and the forearm muscle thickness (cut-off: 1.41 centimeters (cm) men, p=0.017; 0.94 cm women, p=0.041), had great diagnostic accuracy in both sexes.”.
Question 3. Here in the abstract and throughout the whole manuscript, I suggest to substitute gender with sex, as the second refers to the biological sex and it is recommended to discriminate the use of gender and sex as not equal.
Response 3. As the reviewer suggests we have substituted gender with sex throughout the manuscript.
Statistical analysis:
Question 4. In general, this section is too long and clumsy. Be careful to not replicate what is already present in tables or figures in the text.
Response 4. We have improved the statistical analysis as the reviewer recommended (pages 5-6, lines 198-216): “Categorical variables were presented as percentages (%). Continuous variables with normal distribution were presented as the mean ± standard deviation (SD) and variables with non-normal distribution were presented as the median and interquartile range. The normality of the variables was assessed by Kolmogorov-Smirnov test. T‐test and Mann-Whitney U test were used to analyze changes, depending on the nature of the vari-able. Moreover, univariate general linear model was used to assess the differences among variables, adjusted by sex. Also, the effect size of the analysis was calculated.
ROC analysis was performed to assess the diagnostic accuracy of BIA, isokinetic, and ultrasound assessment tools in probable sarcopenic community-dwelling older adults using the area under the curve (AUC) and 95% confidence interval (CI). ROC analysis was significant when AUC is >0.5 and the lower 95% CI value is >0.5 (reference line=0.5) [19]. The diagnostic accuracy based on AUC was the following: ≥0.9 AUC excellent, 0.80-0.89 AUC good, 0.70-0.79 AUC fair, 0.60-0.69 AUC poor, and 0.50-0.59 AUC fail [19]. The Youden’s index was used to determine the cut-off point for each variable based on sensi-bility and specificity to confirm the sarcopenia diagnosis. The higher Youden’s index de-termined the best cut-off point [19].
All statistical analyses were performed using SPSS IBM (Corp. Released 2023. IBM SPSS Statistics for Windows, Version 29.0.1.0 Armonk, NY: IBM Corp.). A p-value of <0.05 was considered significant.”
Question 5. Since the different sample sizes in the two groups (that are not very large themselves), I think that non-parametric testing should be performed to provide more robust statistical support.
Response 5. According to the reviewer's suggestion, we decided to make a non-parametric analysis using the Mann-Whitney U test based on the different small sample sizes in the two groups, as well as the small samples in both groups. Also, we assessed the normality of the variables to express the variables by using mean ± standard deviation (SD) for variables with normal distribution and median and interquartile range for variables with non-normal distribution. Additionally, we have modified the results section updating the p-values.
Question 6. The authors performed different US measurements on the same muscle; as such, I think that this leads to a possible repeated measures error and a correction should be applied.
Response 6. There is no established consensus on the best point of measurement to assess muscle mass with ultrasound. For this reason, we have relied on the points suggested by the scientific bibliography:
- Perkisas, S.; Bastijns, S.; Baudry, S.; Bauer, J.; Beaudart, C.; Beckwée, D.; Cruz-Jentoft, A.; Gasowski, J.; Hobbelen, H.; Jager-Wittenaar, H.; et al. Application of Ultrasound for Muscle Assessment in Sarcopenia: 2020 SARCUS Update. Eur Geriatr Med 2021, 12, 45–59, doi:10.1007/S41999-020-00433-9.
- Perkisas, S.; Baudry, S.; Bauer, J.; Beckwée, D.; De Cock, A.M.; Hobbelen, H.; Jager-Wittenaar, H.; Kasiukiewicz, A.; Landi, F.; Marco, E.; et al. Application of Ultrasound for Muscle Assessment in Sarcopenia: Towards Standardized Measurements. Eur Geriatr Med 2018, 9, 739–757, doi:10.1007/S41999-018-0104-9.
- Meza-Valderrama, D.; Sánchez- Rodríguez, D.; Perkisas, S.; Duran, X.; Bastijns, S.; Dávalos-Yerovi, V.; Da Costa, E.; Marco, E. The Feasibility and Reliability of Measuring Forearm Muscle Thickness by Ultrasound in a Geriatric Inpatient Setting: A Cross-Sectional Pilot Study. BMC Geriatr 2022, 22, doi:10.1186/S12877-022-02811-3.
While in the forearm there is only one measurement point (at 30% proximal between the styloid process of the radius and the insertion of the biceps brachii muscle into the radial tuberosity), in the quadriceps different points could be used to measure the quantity and quality of muscle mass. In order not to underestimate measurement points, it was decided to use both the 50% and the 30% (proximal of the superior border of the patella) between the anterior super iliac spine and the proximal end of the patella. There are no repeated measurements, but rather different measurement points and therefore no correction can be applied.
Question 7. Reporting the effect size of the performed statistical analysis should be recommended, especially due to the limited sample size.
Response 7. As the reviewer suggested, we have calculated the effect size of the different variables and we included the information in the methods section (page 5, line 204: “Also, the effect size of the analysis was calculated.”) and in Tables 2 and 3. Also, according to the results obtained we included this in the limitations section (page 15, lines 433-435): “Third, there is a low effect size due to the small sample size. It would be interesting to improve the analysis with a larger sample.” and we expressed the results obtained with cautious due to the limiting effect size results.
Results:
Question 8. Please, be consistent: men and women, or males and females (Table 1)
Response 8. We have corrected the mistake in Table 1 and decided to use the word “women” throughout the manuscript.
Question 9. Age can be reported with only 1 decimal; the same for body weight. In general, the number of decimals should be based on the precision of the instrument.
Response 9. As the reviewer suggests we have used one decimal in age and weight variables in Table 1.
Question 10. In the notes below the table, it is usually recommended to report also the statistical test used to provide the significance.
Response 10. As the reviewer suggests we have included the statistical test used in the notes below the table (page 6 and line 228, page 7 and 248, page 9 and line 265).
Question 11. Line 249, please report full p-value, not p< 0.05
Response 11. According to the reviewer’s suggestion, we have reported the full p-value of each variable on page 7 and lines 250-251: “extension (PT p<0.001; TW p<0.001; MP p=0.002) and flexion (PT p=0.007; TW p=0.007; MP p=0.011)”.
Discussion:
Question 12. Overall, I think that the authors should be extremely careful with the interpretation of their findings. As they also recognize in the short limitations section, the small sample size, with not equal proportions of sarcopenic and non-sarcopenic individuals, and the unbalanced presence of males and females, does not encourage the performance of statistical analysis that recommends cut-off values. Unfortunately, this is a big bias that must be mentioned, and the whole study needs to be somehow "downgraded" to an exploratory analysis (which should be well stated in the title of the manuscript, abstract and main text). Usually, this kind of studies recruit far more than 100 people to reduce interindividual variability.
Response 12. According to the reviewer’s suggestion we have decided to include the concept “exploratory analysis” in the title (page 1, lines 3-4), the methods of the abstract section (page 1, line 30), the methods section of the manuscript (page 3, line 102), and the discussion section (page 13, line 340).
Moreover, we revised the discussion section about the interpretation of cut-off points. We have decided to be more cautious and understand the proposed cut-off points as a starting point for the assessment of muscle mass quality but more research is needed with a higher sample size. On page 13 and lines 338-340: “Also, some cut-off points for these parameters are suggested to be more representative of the older adult with probable sarcopenia population, as a starting point for an improvement in the confirmation of the diagnosis of sarcopenia.”. In page 13 and line 357: “proposed a possible cut-off point for the phase angle”. In page 13 and lines 365: “a proposal of cut-off points was provided”. In page 13 and line 378: “some possible cut-off points”. In page 14 and line 396: “possible cut-off points”. In page 14 and line 415: “also proposed new possible cut-off points”.
Also, we clarify this in the conclusion section on page 15 and lines 439-442: “suggested cut-off points could be useful to confirm the diagnose of sarcopenia in both sexes in probable sarcopenic individuals, although more research is needed with a higher sample size to confirm this.”.
Question 13. Regarding the US protocol, the authors should be aware that vastus lateralis has been often recommended for the thigh muscles, as it better reflects aging or disuse alterations; in addition, the Ultrasound Sarcopenia Index integrates pennation angle and muscle thickness to limit the bias due to different anthropometrics and sex (Narici et al., 2021; Di Lenarda et al., 2024). I suggest discussing these limitations too according to the literature.
Response 13. According to the reviewer’s suggestion, we have included information about the vastus lateralis and the ultrasound sarcopenia index (pages 13-14, lines 391-394): “Additionally, the scientific evidence also proposed the ultrasound measurement of vastus lateralis as a useful measure to assess the aging effects on muscle, in particular the ultra-sound sarcopenia index which is the ratio between the muscle thickness of the vastus lateralis and the fascicles length [24,25].”.
Also, we have included the 2 references reported by the reviewer and we have updated the reference list.
Reviewer 2 Report
Comments and Suggestions for Authors
I congratulate the authors on developing the present study: New perspectives for low muscle mass quantity/quality assessment in probable sarcopenic older adults: A cross-sectional study. I believe that this paper was well written. It is interesting to insert study limitations, including a low sample size. It is essential to highlight that epidemiological studies could be conducted to confirm or deny these findings. Another point is the transference of these findings daily, where some places do not have the equipment or resources to conduct gold-standard methods. Because of this, using the SARC-F questionnaire, physical fitness tests, and food records (to understand protein consumption) could be related to possible sarcopenia. Congratulation for this work.
Author Response
Nutrients Manuscript: nutrients-2993433
Title: New perspectives for low muscle mass quantity/quality assessment in probable sarcopenic older adults: A cross-sectional study
Authors: Maria Besora-Moreno, Elisabet Llauradó*, Claudia Jiménez-ten Hoevel, Cristina Sepúlveda, Judit Queral, Glòria Bernal, Laura Pérez-Merino, Blanca Alabadi, Yolanda Ortega, Rosa Maria Valls, Rosa Solà*, and Anna Pedret
Dear Editor,
Thank you for your letter and the reviewers’ comments on our manuscript entitled “New perspectives for low muscle mass quantity/quality assessment in probable sarcopenic older adults: A cross-sectional study”.
We appreciate the time and effort that you and the reviewers have dedicated to providing your valuable feedback on our manuscript and we are grateful to the reviewers for their insightful comments on our paper.
Finally, we have been able to address all the issues suggested by reviewers as well as to incorporate into the original manuscript all the proposed changes. Accordingly, we have added the appropriate clarifications in the revised manuscript, which has allowed us to improve the manuscript content. We have highlighted the changes suggested by the reviewers in yellow color within the revised manuscript.
Also, according to the reviewer’s suggestion we changed the title of the manuscript as “New perspectives for low muscle mass quantity/quality assessment in probable sarcopenic older adults: An exploratory analysis study”. Also, we have included a new author that we mistakenly forgot when we submitted the manuscript.
We hope that the revised manuscript meets the quality standard required for publication in Nutrients.
Sincerely,
Elisabet Llauradó and Rosa Solà
Functional Nutrition, Oxidation, and Cardiovascular Diseases Group (NFOC-Salut)
Facultat de Medicina i Ciències de la Salut
Universitat Rovira i Virgili
43201 Reus
Spain
Tel.: (0034) 977758920
E-mail: elisabet.llaurado@urv.cat; rosa.sola@urv.cat
REVIEWER 2
Comment 1. I congratulate the authors on developing the present study: New perspectives for low muscle mass quantity/quality assessment in probable sarcopenic older adults: A cross-sectional study.
Response 1. Thank you for your appreciation.
Comment 2. I believe that this paper was well written. It is interesting to insert study limitations, including a low sample size. It is essential to highlight that epidemiological studies could be conducted to confirm or deny these findings.
Response 2. We have improved the limitation section on page 15 and lines 433-435: “Third, there is a low effect size due to the small sample size. It would be interesting to im-prove the analysis with a larger sample.”.
Comment 3. Another point is the transference of these findings daily, where some places do not have the equipment or resources to conduct gold-standard methods. Because of this, using the SARC-F questionnaire, physical fitness tests, and food records (to understand protein consumption) could be related to possible sarcopenia. Congratulation for this work.
Response 3: We appreciate your comments and observations. As we said in discussion section (page 14 and lines 418-421): “With the evaluation of these parameters, the complaints of the individuals in the present study on the functional domain will be understood, such as climbing stairs; representing a useful translation strategy into clinical practice.” it is important to use other tools more available in clinical practice such as ultrasound.